# Ivermectin as an Alternative Anticancer Agent: A Review of Its Chemical Properties and Therapeutic Potential

**DOI:** 10.3390/ph18101459

**Published:** 2025-09-28

**Authors:** Kimberly Naula Robalino, Oscar Vivanco-Galván, Juan Carlos Romero-Benavides, Yuliana Jiménez-Gaona

**Affiliations:** 1Departamento de Química y Ciencias Exactas, Universidad Técnica Particular de Loja, San Cayetano Alto s/n, Loja 1101608, Ecuador; klnaula1@utpl.edu.ec (K.N.R.); jcromerob@utpl.edu.ec (J.C.R.-B.); 2Departamento de Ciencias Biológicas y Agropecuarias, Universidad Técnica Particular de Loja, San Cayetano Alto s/n, Loja 1101608, Ecuador; oavivanco@utpl.edu.ec

**Keywords:** anticancer, ivermectin, chemical properties, therapeutic potential

## Abstract

**Background:** Ivermectin has recently garnered significant scientific attention for its potential anticancer properties. **Objective:** This research aims a comprehensive literature review to evaluate IVM’s chemical characteristics and assess its applicability as an alternative therapeutic strategy in oncology. **Methods:** The methodology involved a systematic search and critical appraisal of data from peer-reviewed scientific databases, focusing on structural analyses, such as nuclear magnetic resonance (NMR), crystallography, and in silico modeling, as well as preclinical experimental studies. **Results:** The review highlights IVM’s distinct physicochemical profile, including high lipophilicity, poor aqueous solubility, and moderate acid stability, which collectively affect its bioavailability and pharmacokinetic behavior. Mechanistically, IVM has been shown to modulate multiple oncogenic signaling pathways, including Wnt/β-catenin, PI3K/Akt/mTOR, and STAT3. These interactions contribute to the induction of apoptosis, inhibition of tumor cell proliferation, and modulation of the tumor microenvironment across a range of malignancies. Despite encouraging preclinical evidence, clinical validation remains limited. **Conclusions:** Further investigation is needed to optimize IVM’s formulation for enhanced solubility and targeted delivery, as well as to design robust clinical trials assessing its safety and efficacy in oncology settings. This review provides a foundational framework for future interdisciplinary research on drug repurposing and highlights the potential of IVM as a cost-effective and accessible adjunct or alternative to modern cancer therapy.

## 1. Introduction

Cancer remains one of the most significant global health challenges and a leading cause of mortality worldwide [1]. Despite substantial advances in conventional oncological treatments, such as chemotherapy and radiotherapy, these options often present major limitations, including severe side effects and the development of tumor resistance [2]. In this context, drug repurposing has emerged as a promising strategy in cancer research. Ivermectin (IVM), a widely used antiparasitic agent in both veterinary and human medicine, has recently garnered significant scientific attention due to its potential anticancer activity [3,4,5,6].

Recent studies have provided compelling evidence regarding IVM’s mechanisms of action in cancer cells, demonstrating its capacity to modulate multiple oncogenic signaling pathways, including Wnt/β-catenin, PI3K/Akt/mTOR, and STAT3. These interactions contribute to apoptosis induction, inhibition of tumor cell proliferation, and modulation of the tumor microenvironment across a range of malignancies [7]. Despite this encouraging preclinical evidence, clinical validation remains limited [8,9]. For instance, a rural study conducted in Loja, Ecuador, revealed that while 18.75% of surveyed cancer patients reported using IVM as an alternative therapy [9], most physicians warned about the lack of clinical evidence supporting its oncological application [10].

Yuwen et al. [11] published a review of IVM use in cancer patients; the results show that only 26 publications met the criteria involving cancer patients treated with ivermectin for parasitic infections. These individual-level data suggested that ivermectin was generally well tolerated in these patients—even those undergoing chemotherapy. Thus, IVM has emerged as a promising candidate for repurposing in oncology; however, it has not yet been clinically evaluated in cancer patients and remains primarily recognized as a well-established antiparasitic agent [3].

Its pharmacology, safety, and toxicity have been thoroughly investigated in both humans and animals, with data indicating a favorable safety profile and very low toxicity in humans [12]. Preclinical studies provide encouraging evidence of anticancer potential, showing that IVM can reduce tumor volume by more than 50% in murine models, even at doses lower than the maximum safe limits defined for humans [13]. While these findings are compelling, they should be interpreted with caution: results obtained in animal models may not directly translate to clinical efficacy. Rigorous clinical trials are therefore essential to determine whether the antitumor effects observed in preclinical studies can be safely and effectively reproduced in cancer patients.

Thus, based on the previously describe the novelty of this research lies in exploring the therapeutic potential of a well-known, accessible, and economically viable drug [14], with a documented safety profile and broad availability [1]. While IVM’s antitumor potential has been recognized [9], there is a notable absence of recent systematic reviews specifically addressing its chemical properties and how these influence its therapeutic potential in oncology. This review seeks to fill this gap by providing a foundational framework for future interdisciplinary research and highlighting IVM’s value as a cost-effective and accessible adjunct or alternative to modern cancer therapy.

The main objective of this review is to provide a comprehensive analysis of ivermectin (IVM), focusing on its chemical characteristics (Figure 1) and potential applicability as an alternative therapeutic strategy in oncology. To achieve this, we conducted a systematic search of relevant scientific databases and structured our aims as follows: (i) to describe the chemical structure of IVM, including its physicochemical properties, stability, and solubility, and to discuss how these features influence its biological activity; (ii) to analyze studies examining how IVM interacts with key molecules and signaling pathways in cancer cells, including receptors, enzymes, and metabolic processes; and (iii) to evaluate the current evidence regarding its efficacy in different cancer cell lines and preclinical models, particularly its ability to inhibit cell proliferation and induce apoptosis.

### Related Work

In recent years, numerous studies have explored the potential antitumor properties of IVM, revealing its multifaceted role in cancer inhibition. Originally developed as an antiparasitic agent, IVM has attracted interest for its ability to interfere with various molecular and cellular pathways critical to cancer progression.

Several studies have identified IVM’s capacity to modulate key cancer-related signaling pathways. For example, Rabben et al. [6] found that IVM inhibits the Wnt/β-catenin signaling pathway in gastric cancer cells, leading to suppressed tumor growth and metastasis. Similarly, Lee et al. [10] demonstrated that IVM disrupts the PI3K/AKT/mTOR pathway in pancreatic cancer, resulting in increased apoptosis and reduced cell proliferation. Additionally, the drug has shown the ability to modulate STAT3, EGFR, and other crucial pathways across multiple cancer types [13,14].

Experimental evidence has also revealed IVM’s inhibitory effects on various types of cancer cells. Zhou et al. [15] reported that IVM suppresses colorectal cancer cell growth in a dose-dependent manner, promotes apoptosis via caspase activation, and upregulates pro-apoptotic proteins while reducing anti-apoptotic markers. In ovarian and prostate cancers, IVM has demonstrated the ability to inhibit cell growth, modulate gene expression, and alter DNA repair mechanisms [16,17]. Beyond conventional tumor cells, IVM also targets cancer stem cells (CSCs), which are often implicated in treatment resistance and recurrence. Dominguez et al. [5] showed that IVM selectively suppresses CSCs in breast cancer models and downregulates genes associated with cellular stemness, suggesting a valuable role in long-term therapeutic strategies.

Combination therapies involving IVM have also yielded promising results. Draganov et al. [18] demonstrated that combining IVM with immune checkpoint inhibitors, such as anti-PD1 antibodies, enhances antitumor immune responses and induces complete tumor regression in breast cancer models. Similarly, Nunes et al. [19] reported synergistic effects when IVM was paired with paclitaxel in drug-resistant ovarian cancer cell lines.

Despite encouraging preclinical outcomes, many studies have been conducted in vitro or in animal models. The translation of these findings to clinical practice remains a significant challenge. Jiménez et al. [9], in a rural observational study in Ecuador, noted that 18.75% of surveyed cancer patients used IVM as an alternative therapy. While some reported symptomatic improvements, oncologists emphasized the lack of validated clinical evidence supporting its use in oncology.

Collectively, the literature supports the idea that IVM holds considerable promise as an anticancer agent, acting through diverse pathways and mechanisms. However, robust clinical trials are essential to verify its safety, efficacy, and optimal dosing in human populations.

## 2. Results

The experimental results are described according to the objectives established: (i) Findings on the chemical properties of IVM, (ii) Effects of IVM on target molecules and signaling pathways in cancer cells, and (iii) Review of the clinical applicability of IVM in oncology.

### 2.1. Chemical Properties of IVM

Table 1 and Table 2 summarize the most relevant studies on (i) the chemical properties of IVM and (ii) its main application involving target molecules and signaling pathways in cancer cells, respectively.

Figure 2 shows the level of publications by the author during the first bibliometric search about “Physicochemical and pharmacokinetic properties of IVM” according to the type of study.

### 2.2. IVM Effects Target Molecules and Signaling Pathways in Cancer Cells

Table 2 and Figure 3 detail the main findings on the application of IVM on target molecules and signaling pathways in cancer cells.

Figure 3 shows the percentage of inhibition of specific molecular pathways in different types of cancer after treatment with IVM.

Table 3 provides an overview of the current preclinical and emerging clinical data regarding ivermectin’s potential role as an anticancer agent. The table highlights the spectrum of tumor types in which IVM has demonstrated activity, including colorectal, breast, lung, and melanoma models, and outlines the principal mechanisms reported, such as apoptosis induction, mitochondrial dysfunction, oxidative stress, and modulation of signaling pathways like NF-κB and mTOR/STAT3. Figure 4 shows the effects of ivermectin (efficacy) vs. cancer type and the inhibition of cell proliferation and induction of apoptosis in in vitro models of different types of cancer.

## 3. Discussion

This literature review provides an integrated perspective on the chemical characteristics of ivermectin (IVM), its mechanisms of action in tumor cells, and its potential applications as an anticancer agent. The analysis highlights key advances in understanding its physicochemical properties, its interaction with molecular pathways relevant to oncology, and the ongoing challenges for clinical translation [33,34].

Recent studies (Table 2) have clarified that IVM possesses a stable macrocyclic structure with B1a and B1b isoforms, and a high log P (~5.8), reflecting its lipophilicity. Structural elucidation techniques such as X-ray crystallography and thermal analysis [35,36,37,38] have been fundamental in characterizing its interactions with intracellular targets. A major limitation remains its poor aqueous solubility; however, strategies including co-crystallization with lipophilic excipients [24] and incorporation into silica- or polymer-based nanostructures [36] have significantly enhanced dissolution and bioavailability. Likewise, stability concerns due to UV radiation and pH sensitivity [25] underscore the importance of protective formulations. As summarized in Figure 2 advances in controlled-release technologies and solid-state modifications provide promising avenues to simultaneously improve solubility, stability, and therapeutic performance.

With respect to biological activity, evidence synthesized in Table 2 indicates that IVM interferes with multiple oncogenic pathways, including WNT/TGF-β, PAK1/STAT3, YAP1, Akt/mTOR, and Wnt/β-catenin. These effects are consistent with reports of inhibited proliferation and apoptosis induction [39], exemplified by breast cancer models in which IVM blocked WNT/TGF-β signaling (Figure 3). Nonetheless, variability in response across tumor types [10,34] suggests that therapeutic efficacy strongly depends on molecular context, emphasizing the need for predictive biomarkers to guide personalized applications.

Clinical viability analyses (Table 2 and Figure 4) further indicate that IVM exhibits potential activity against lung, colon, ovarian, melanoma, and pancreatic cancers [16,17,31,32,33,40]. Both in vitro and in vivo studies support its capacity to inhibit tumor growth, activate caspases, and modulate critical signaling pathways such as PI3K/Akt and STAT3 [3]. Still, the real-world use of IVM in oncology remains anecdotal, as shown by Jiménez et al. [9] in rural Loja, where patients reported off-label use despite the lack of clinical validation.

Overall, the reviewed evidence [41,42,43,44] underscores IVM’s promise as an anticancer agent while highlighting significant gaps in formulation science, biomarker discovery, and translational research. Addressing these challenges will be essential to defining its role in future oncological therapeutics.

## 4. Methodology

The methodology involved a systematic search and critical appraisal of data from peer-reviewed scientific databases such as PubMed, Scopus, Google Scholar, and ScienceDirect, focusing on structural analyses, including nuclear magnetic resonance (NMR), crystallography, in silico modeling, and preclinical experimental studies. The bibliometric tool VOSviewer v 1.6.20 was also employed to identify research trends and collaborations.

The general systematic review flowchart is divided into five sections, as shown in Figure 5, and the search was mainly focused on two main categories: (i) the chemical properties of IVM and (ii) its potential applicability in alternative cancer treatments based on the most recent literature. Also, the articles were critically appraised to determine their robustness.

### 4.1. Relevant Studies Selection

The literature review was conducted using the recommendations given by Khan et al. [19], the methodology proposed by Torres-Carrión [45], and the protocol proposed by Page et al. [46]. The Preferred Reporting Items for Systematic Reviews and Meta-Analyses (PRISMA) flowchart is shown in Figure 6.

Inclusion Criteria:Original experimental studies, excluding reviews, individual case reports, and bibliographic works.Research applying highly complex analytical techniques for molecular characterization and developing rigorous preclinical studies.Studies focused on the detailed analysis of the IVM’s chemical properties, as well as its potential anticancer effect.Research exploring the molecular mechanisms involved in oncological contexts.Studies using standardized methodologies to evaluate both the antitumor activity and cytotoxicity of IVM.Publications from 2014 to 2024, avoiding duplication or redundant records.

Exclusion Criteria:Studies focused exclusively on the antiparasitic activity of IVM.Research conducted solely in silico without experimental validation.Studies that do not provide data on molecular mechanisms or chemical characterization.Research does not specifically evaluate anticancer effects.

#### 4.1.1. Initial Search

The first literature search was conducted between October 2023 and January 2024, using a combined strategy with the key terms (“ivermectin,” “cancer treatment,” and “chemical properties”) and Boolean operators (AND/OR) applied in specialized scientific databases, including PubMed, ScienceDirect, Scopus, Google Scholar, and Web of Science. The search strategy was further refined by incorporating additional terms such as antineoplastic activity, molecular mechanisms, preclinical studies, drug repurposing, cell signaling, apoptosis, antiproliferative effects, and cancer therapy.

This initial exploration generated a large volume of publications, with considerable overlap between databases and several articles unrelated to the research objective. Nevertheless, this phase was essential to obtaining a broad and well-grounded overview of the research landscape. It also revealed a notable gap: the absence of recent systematic reviews focused specifically on the chemical properties of IVM in the context of oncology treatment.

Based on a comparative evaluation of the databases, considering both scope and degree of specificity, Web of Science was excluded from the final systematic search. This decision was made because the platform did not provide additional relevant studies beyond those already identified in PubMed, Scopus, Google Scholar, and ScienceDirect, which offered more comprehensive and relevant coverage for the objectives of this research.

#### 4.1.2. Systematic Search

The combination of terms that yielded the most relevant results in the databases is presented in Table 4. The table lists the Boolean equations used during the systematic review, structured around three principal axes: cancer-related treatments, the chemical properties of IVM, and its characteristics in oncology settings.

The search yielded 1471 articles, distributed as follows: 287 in PubMed, 412 in ScienceDirect, 523 in Google Scholar, and 195 in Scopus. Before proceeding with the selection and analysis of the collected papers, inclusion and exclusion criteria were applied to ensure thematic relevance and methodological quality.

### 4.2. Scopus Data Collection

The previously defined keywords were also used to generate a search in the Scopus database (https://www.scopus.com). After applying the inclusion and exclusion criteria, a filtered CSV file was generated and used as input in VOSviewer [47] for constructing and visualizing bibliometric networks. These networks include journals, researchers, or individual publications, and can be built based on citation, bibliographic coupling, co-citation, or co-authorship relations.

#### 4.2.1. VOSViewer Bibliometric Map

A network map was created in VOSviewer using several items (networks of co-authorship, citation, and bibliographic linkage) with the options “Create a map based on bibliographic data” and “Read data from bibliographic database” (Figure 7 and Figure 8). These maps were also generated from the Web of Science, Scopus, Dimensions, and PubMed databases.

The five variables used to create the heat maps are described in Table 5. By default, VOSviewer applies the first category (co-authorship analysis), but thresholds must be defined for the number of documents and citations per author. In this study, we set the document threshold to 1 and the author citation threshold to 5; with these parameters, the resulting network comprised 102 authors.

From the bibliometric analysis, a scientific collaboration network can be identified, composed of leading researchers dedicated to studying IVM and its potential as an anticancer agent. This network presents an interrelated configuration in which each node, representing an author, varies in size according to the number of citations received, while the thickness of the connecting lines reflects the intensity of collaborative links.

The spatial distribution of the map clearly reveals three research clusters or subgroups, each organized around highly relevant academic figures (Figure 8a–c). This topological structure highlights the citation and cooperation flows between the most influential authors in the field and provides a clear, visually interpretable representation of how scientific knowledge is organized around the study of IVM in oncology. The central position of certain researchers within the network suggests that they have played a key role in advancing this line of research.

The bibliometric analysis performed with VOSviewer reveals a highly organized research structure around IVM and its anticancer applications, evidenced by three interconnected and differentiated nodes (Figure 8a). The red node focuses on clinical and pharmacological aspects, including human research and the drug’s mechanisms of action, and highlights terms such as “drug mechanism” and “tumor necrosis factor.” The green node addresses experimental studies, emphasizing drug repositioning and antineoplastic activity, while the blue cluster concentrates on preclinical research, including animal experiments and cell viability studies. The central position of the term “ivermectin” and the density of connections between nodes reveal a strong interdisciplinary research base, evolving from basic studies in animal models to potential clinical applications supported by scientific evidence across multiple levels of biomedical research.

Bibliometric analysis using VOSviewer further demonstrates a multifaceted research structure around ivermectin, organized into three distinctive and interrelated nodes (Figure 8b). The red node, focused on molecular and preclinical studies, is characterized by terms such as “molecular docking” and “binding affinity,” which establish the mechanistic underpinnings of the research. The green node emphasizes clinical aspects, including drug efficacy and safety, while the blue node situates IVM within the broader pharmacological landscape, emphasizing its relationship with other drugs such as remdesivir and hydroxychloroquine.

The central and dominant position of the term “ivermectin” on the map, together with the high density of connections between nodes, indicates a mature and cohesive field of research that integrates basic research with clinical applications, demonstrating systematic and comprehensive development of this area (Figure 8c). The red node highlights clinical and pharmacological aspects, focusing on human research and the drug’s mechanisms of action, with terms such as “drug mechanism” and “tumor necrosis factor.”

Since 2015, the evolution of the field has been evident in the chromatic gradation of the data, which reveals a gradual shift from primary studies in animal models to more recent research addressing chemical properties and antioxidant activity. The interconnections between clusters, visible in the density and arrangement of nodes, demonstrate a strong articulation between basic science and clinical application, reflecting a sustained and coherent advancement of knowledge in this area.

#### 4.2.2. Statistical Analysis with AI Tools

Statistical analysis of the data obtained from the Scopus database (csv) was performed using the tools DeepSeek-V3.1 (https://chat.deepseek.com/) and ChatGPT 5 (https://chatgpt.com) to generate statistics and visualizations on research trends related to the use of ivermectin in medicine, its properties, and oncological effects.

## 5. Conclusions

This review highlights the growing body of evidence supporting ivermectin (IVM) as a potential anticancer agent. Its chemical characteristics, such as high molecular weight, lipophilicity, and low water solubility pose significant challenges for systemic application; however, advances in drug delivery strategies, including nanoencapsulation and co-crystallization, show promise in overcoming these barriers.

Across diverse tumor models, IVM demonstrates consistent antitumor activity through the induction of apoptosis, oxidative stress, mitochondrial dysfunction, and inhibition of key signaling pathways, including NF-κB, mTOR/STAT3, and importin β. In addition, its immunomodulatory effects suggest potential synergy with immunotherapies, particularly in aggressive cancers such as TNBC and melanoma. Preclinical findings also indicate enhanced efficacy when IVM is combined with conventional chemotherapeutic agents.

Nonetheless, the disparity between effective in vitro concentrations and achievable plasma levels in humans underscores the need for translational research. Future work should focus on optimizing formulation and delivery systems, while rigorous clinical trials remain essential to establish the safety, efficacy, and therapeutic window of IVM in oncology.

## 6. Limitations and Future Work

The review of the consulted literature has revealed that ivermectin (IVM), widely recognized for its antiparasitic use, also possesses physicochemical properties and mechanisms of action with therapeutic potential in the field of oncological treatment. However, its adoption as an effective antineoplastic drug requires overcoming several technical and scientific obstacles, as well as opening new areas of research to support the consolidation and expansion of its clinical use.

A fundamental and indispensable step is validation through clinical trials. However, most studies on the antitumor activity of ivermectin have been conducted under in vitro conditions or in animal models, which limits the ability to extrapolate their results to clinical practice. Urgent action is needed in the design and execution of controlled clinical studies that rigorously evaluate its safety profile, efficacy, and optimal dosing in patients with various types of cancer.

At the same time, it is crucial to advance the optimization of its pharmaceutical formulations. Due to its poor solubility in water and limited bioavailability when taken orally, new formulations, such as nanoparticles or controlled release platforms, must be investigated to improve its absorption and facilitate more efficient distribution to tumor tissues.

An additional priority is to deepen the understanding of its molecular mechanisms of action. While IVM’s ability to induce both apoptosis and autophagy in malignant cells have been documented, the cellular processes and pathways involved have not yet been fully clarified. It is essential to understand how it interacts with key signaling cascades like WNT/β-catenin or PI3K/Akt/mTOR, and how these interactions vary according to the type of neoplasm.

Likewise, the analysis of combination therapies should receive particular attention. The possibility of using ivermectin in conjunction with conventional chemotherapy agents or targeted treatments represents a promising avenue for research. Future studies could focus on discovering pharmacological synergies that enhance therapeutic effectiveness while simultaneously reducing the toxicity associated with standard treatments.

Moreover, it is recommended to direct some efforts toward specific oncological pathologies. Although antitumor effects have already been reported in cancers such as breast, colon, and lung, evaluating IVM’s action in less explored tumors, particularly those with limited therapeutic options—could open new clinical perspectives.

The field of pharmacogenomics represents another strategic dimension. The individual response to ivermectin may be influenced by specific genetic variants, so studies focusing on this aspect would allow the identification of predictive biomarkers for both efficacy and toxicity. This information would be crucial to advancing towards personalized medicine models that are safer and tailored to the genetic characteristics of each patient [10].

## Figures and Tables

**Figure 1 pharmaceuticals-18-01459-f001:**
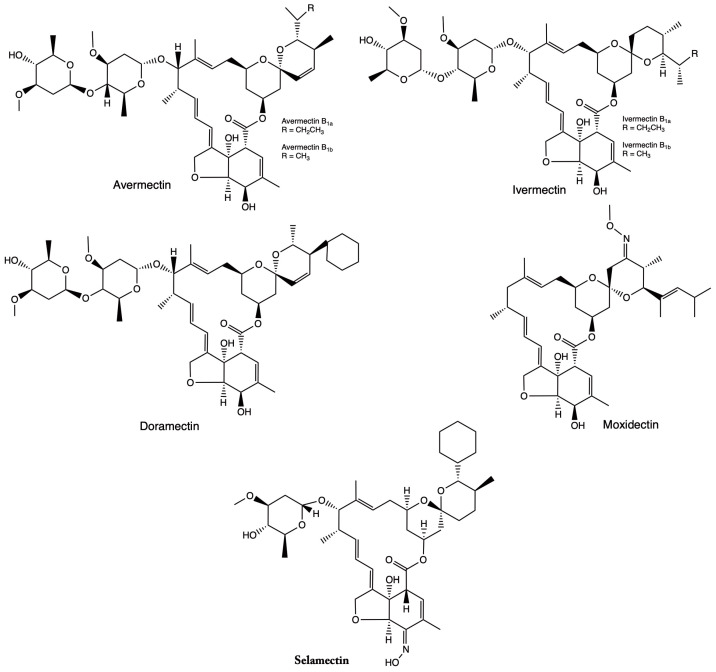
Chemical structure of ivermectin (IVM) and some of its derivatives.

**Figure 2 pharmaceuticals-18-01459-f002:**
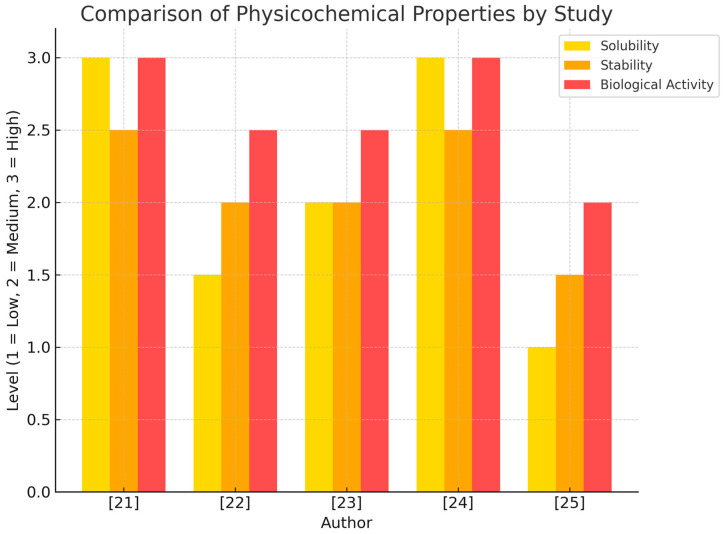
Qualitative evaluation of physicochemical and pharmacokinetic properties of IVM according to study type.

**Figure 3 pharmaceuticals-18-01459-f003:**
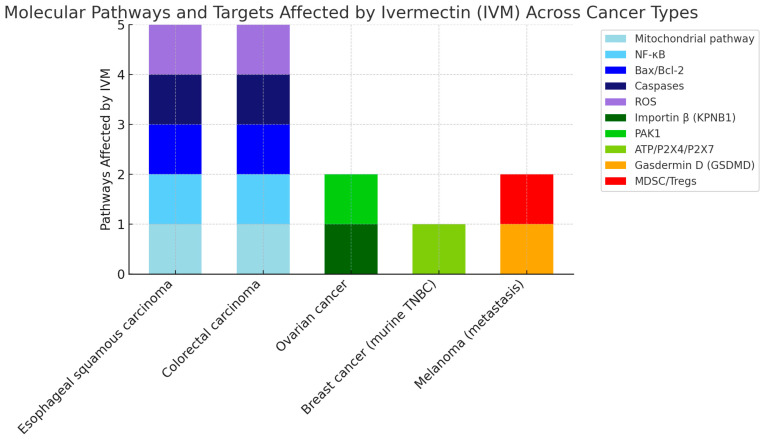
This figure summarizes the molecular mechanisms influenced by Ivermectin across several cancer models. Each stacked bar represents the number and type of pathways affected in a given cancer type, with colors corresponding to specific molecular targets: mitochondrial pathway (light blue), NF-κB (cyan), Bax/Bcl-2 (blue), caspases (dark blue), ROS (purple), importin β (KPNB1, light green), PAK1 (green), ATP/P2X4/P2X7 (yellow-green), gasdermin D (GSDMD, orange), and MDSC/Tregs (red). Esophageal squamous carcinoma and colorectal carcinoma show modulation of five distinct pathways, suggesting a broad mechanism of action. In contrast, ovarian cancer, murine triple-negative breast cancer (TNBC), and metastatic melanoma involve fewer pathways, but still reflect targeted effects relevant to apoptosis, immune regulation, and tumor progression. This visualization highlights the pleiotropic and context-dependent actions of Ivermectin, supporting its potential role as a multi-target anticancer agent.

**Figure 4 pharmaceuticals-18-01459-f004:**
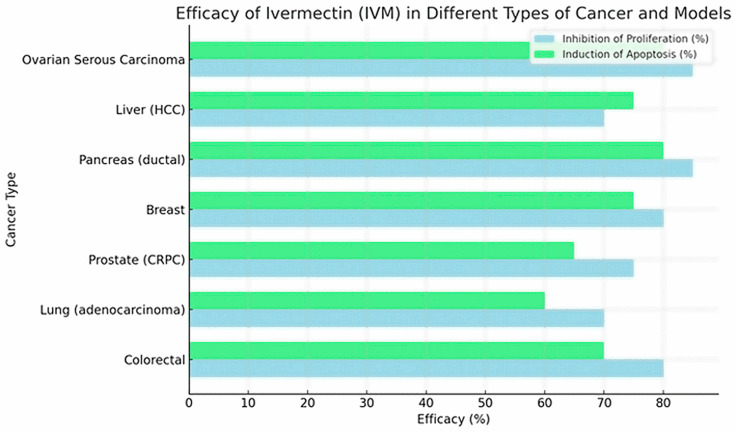
Effects of IVM on Cell Proliferation and Apoptosis in Cancer Models.

**Figure 5 pharmaceuticals-18-01459-f005:**
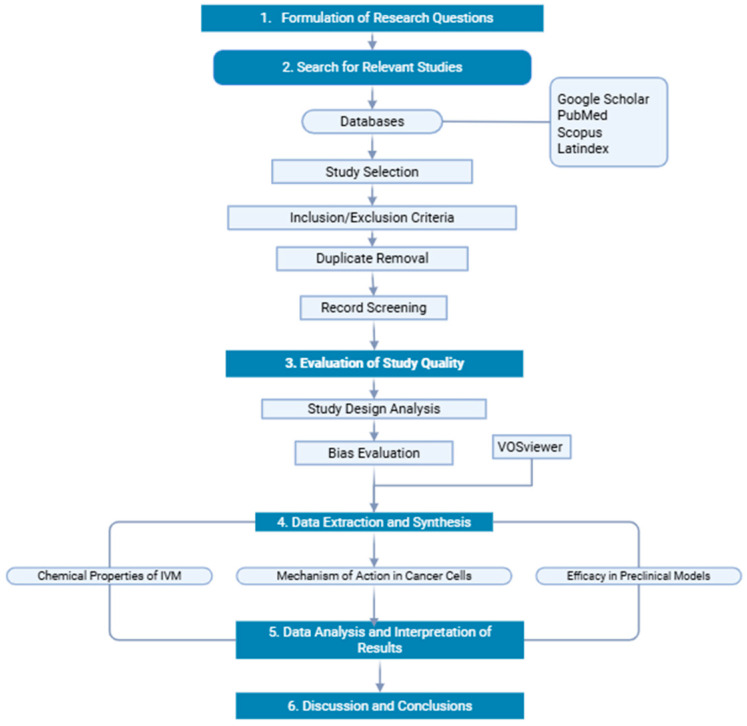
The diagram represents the systematic methodological process. Created by Biorender https://app.biorender.com, accessed on 1 March 2025.

**Figure 6 pharmaceuticals-18-01459-f006:**
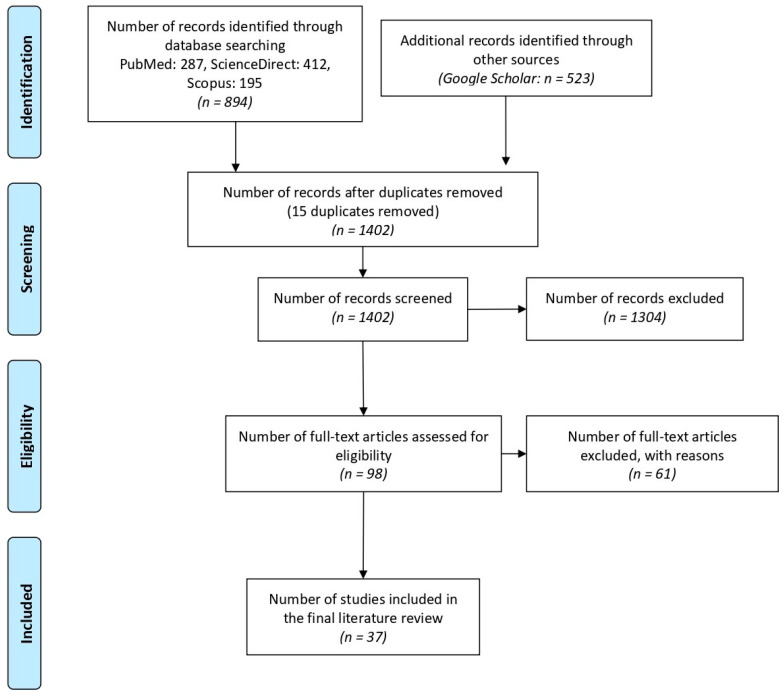
The diagram provides a structured overview of the article selection process, beginning with the initial search in databases such as PubMed, ScienceDirect, Google Scholar, and Scopus, and culminating in the inclusion of 37 studies in the meta-analysis.

**Figure 7 pharmaceuticals-18-01459-f007:**
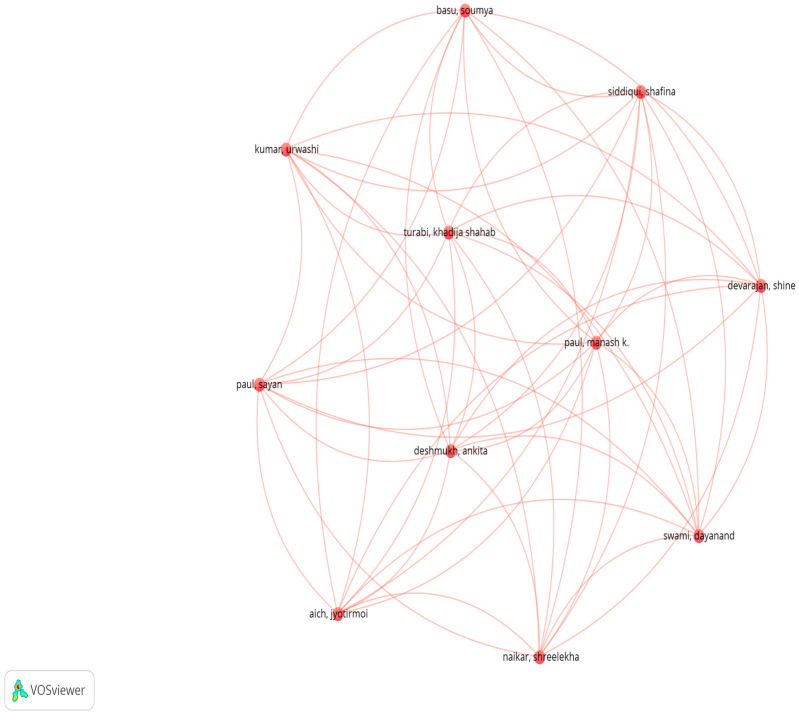
VOSViewer bibliometric analysis by authors.

**Figure 8 pharmaceuticals-18-01459-f008:**
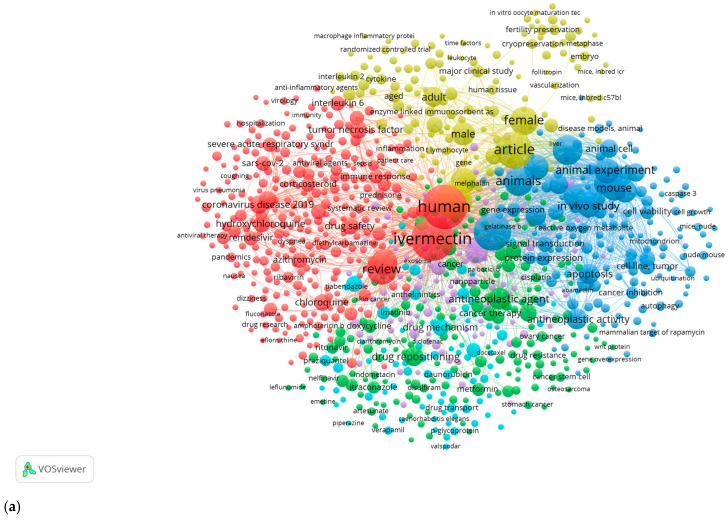
Bibliometric analysis of keywords in VOSviewer. (**a**) Thematic distribution of research on humans, animals, molecular mechanisms, and clinical variables; (**b**) Connection between molecular, clinical, and pharmacological studies (**c**) Time evolution and interdisciplinary relationships in the study of ivermectin.

**Table 1 pharmaceuticals-18-01459-t001:** Main findings on the chemical properties of IVM and its applications in different structures. (Table Notes: ↑: increase).

Studying Type /Reference	Chemical Structure	Physicochemical Properties	Stability	Solubility	Influence on Biological Activity	Conclusion
Crystallography, thermal study [20]	Formula C_48_H_74_O_14_ (B1a) and C_47_H_72_O_14_ (B1b); macrocycle structure with disaccharides	Molecular weight ~875 Da; high log P; melting point ~157 °C; aqueous solubility ~1 µg/mL	Polymorphism and pseudopolymorphism affect stability depending on pH and temperature	Low in water, higher in organic solvents	Crystalline stability impacts bioavailability	Polymorphism affects solubility and stability of solid ivermectin
Co-crystals with Span^®^ 60 (Sigma-Aldrich, MO, USA) [21]	Macrocycle B1a/B1b in solid co-crystalline matrix	Log P ~3.0–5.8; molecular weight 875 Da; solubility improves in oils (+169%)	Stable at room temperature and in thermal processes (DSC/TGA)	↑solubility in oils due to co-crystals	Better penetration in lipid formulations	Co-crystallization improves solubility without altering the base structure
Experimental photostability [22]	Ivermectin B1a	Log P ~5.8; photolabile under UVA/UVC; susceptible to photodegradation	Degrades rapidly at extreme pH or under radiation	Low in water; stable in non-polar solvents	Photodegradation reduces biological efficacy	Requires light protection to maintain potency
Nanoencapsulation (silica and polymers) [23]	Lactone macrocycle B1a/B1b without alteration	BCS Class II; log P 5.83; solubility ~4 µg/mL	Stable in nano-containers	Aqueous solubility increases with nanoencapsulation	Higher bioavailability due to better release	Nanocarriers improve dissolution and activity
Crystallography [24]	Pseudopolymorphic variants (ethanol, GVL, MTBE)	—	Pseudopolymorphs are stable at different temperatures	Influenced by solvent in the crystal lattice	Crystalline forms with different release rates	Controlling solid form is key to standardizing activity
Experimental (environmental stability) [25]	Structure B1a/B1b	Optimal pH ~6.3; unstable under extremes (temperature, light)	Photolysis in a few hours; hydrolysis at extreme pH	No quantitative data; low in water	Reduces action time and environmental efficacy	Storage control is essential to maintain effectiveness

**Table 2 pharmaceuticals-18-01459-t002:** The main findings are on the application of IVM on target molecules and signaling pathways in different cell lines. (Table Notes: ↑: increase; ↓: decrease; →: stable).

Type of Cancer/ Reference	Cell Lines Studied	Target Molecules/Pathways	Mechanism of Action	IVM Dose	Effects on Cancer Cells	Toxicity/IC50 in Normal Cells	Main Findings (Efficacy/Safety)
Esophageal Squamous Cell Carcinoma [26]	KYSE-30, KYSE-70 (tumor); NE-3 (normal)	Mitochondrial pathway and NF-κB: ↑ROS, ↑Bax/↓Bcl-2, caspases	IVM induces mitochondrial dysfunction (↓ψm, ↓ATP) with ↑ROS, inhibits NF-κB (↓p-p65), increases Bax/Bcl-2 and activates caspases 9/3 → apoptosis	IC50 ≈6 μM (KYSE-30), =10 μM (KYSE-70) (estimated)	↓Viability and proliferation, G1 arrest, ↑apoptosis (nuclear fragmentation, more Bax and caspases)	NE-3 normal cells: no toxicity up to ~15 μM; mild effect (~20% inhibition) at 20 μM	IVM (~10 μM) kills ESCC cells via mitochondrial apoptosis, without damaging normal cells at moderate doses
Colorectal Cancer [15]	SW480, SW1116	Mitochondrial pathway and ROS: Bax/Bcl-2, caspases	↑Total/mitochondrial ROS → mitochondrial damage → ↑Bax/↓Bcl-2, ↑caspase-3/7 → apoptosis; also, S phase arrest at low doses	2.5–20 μM (e.g., 2.5–5 μM induced S phase arrest)	↓Dose-dependent viability; ↑caspase-3/7, ↑cell apoptosis (markers: cleaved PARP)	Not evaluated in this study; focus on tumor cells	IVM suppresses CRC growth via ROS-mediated apoptosis (NAC reverses the effect)
Ovarian Cancer [27]	COV-318, OVCAR-5, CAOV-3, A2780, TOV-21G, SKOV-3	Importin β (KPNB1)/PAK1 (indirect)	Alone: modulates KPNB1 → cycle arrest. In combination synergizes with pitavastatin, ↑apoptosis (caspase-3/7)	IVM alone: IC50 ~10–20 μM; in synergy tested ~20 μM fixed	IVM alone: moderate growth inhibition; combined: greater viability reduction (CI~0.6 in COV-318) and much ↑apoptosis (caspase-3/7 (2–4 fold)	IC50 ≫ safe plasma levels (e.g., 10–20 μM vs. ~0.05–0.3 μM possible in vivo)	IVM enhances pitavastatin in ovarian cancer, suggesting combination therapy; however, required in vitro concentrations are very high compared to safe human levels
Breast Cancer (Murine TNBC) [18]	4T1 (mouse cell line)	ATP/P2X4/P2X7 axis, ICD mediators (ATP, HMGB1), MDSC/Tregs	IVM modulates ATP/P2X4/P2X7 channels → induces immunogenic cell death (releases ATP, HMGB1), selectively depletes MDSCs and Tregs → ↑CD8+ T infiltration and ↑Teff/Treg	12 μM (used ex vivo in 4T1 cells for vaccine)	Alone: no noticeable antitumor effect. With anti-PD-1: synergy limits tumor growth (*p* = 0.03) and increases complete remissions. ↑immune response against rechallenge	Not reported (immunocompetent mouse model)	IVM acts as an immune modulator, “converts” cold tumors into hot ones; alone it doesn’t reduce tumors, but strongly boosts anti-PD-1 therapy
Melanoma (Metastasis) [28]	Neutrophils (mouse); B16F10 (murine)	Gasdermin D (GSDMD), NETs, MDSC	IVM binds to GSDMD (Kd ≈ 0.268 μM) and blocks its oligomerization → inhibits NET formation. Reduces infiltrated MDSCs and ↑CD8+ T in the lung	Kd ~0.268 μM (affinity to GSDMD); 5 μM did not affect B16F10 in 48h	No effect on primary tumor; ↓significant lung metastasis; ↓ctDNA of NETs in serum; ↑CD8+ in metastasis	B16F10: IC50 much higher than 5 μM; no viability reduction at 5 μM	IVM did not reduce the primary tumor, but stopped lung metastases in mice (via blocking NETs/GSDMD), suggesting potential anti-metastatic activity

**Table 3 pharmaceuticals-18-01459-t003:** Clinical applicability of IVM in oncology. (Table Notes: ↑: increase; ↓: decrease; MMP-Δψm: Mitochondrial membrane potential).

Reference	Cancer Type	Model In Vitro/In Vivo	Treatment IVM Concentration	Main Findings (Efficacy)	Combination	Dose (I.V.)	Molecular Pathway/Target	Exposure Time	Toxicity/Selectivity
[15]	Colorectal	In vitro: SW480, SW1116 (human colorectal cancer).	IVM in increasing dilutions (0–30 µM).	Dose-dependent inhibition of viability and proliferation in SW480/SW1116; ↑apoptosis (Annexin-V+, ↑caspase-3/7 activity); ↑proportion of cells in S (S-phase arrest). Decreases Bcl-2; ↑Bax, ↑cleaved PARP.	--	–	↑ROS; mitochondrial apoptotic pathway (↑Bax, caspase-3); ↓Bcl-2; ↑cleaved PARP.	6–36 h (depending on assay)	Not reported
[29]	Lung (adenocarcinoma)	In vitro: LUAD lines (e.g., A549 or H1975; non-specific); In vivo: LUAD xenografts in nude mice.	IVM (exact µM not detailed in summary; usually 5–20 µM) in culture; in mice 10 mg/kg i.p., 3×/week.	Marked inhibition of colony formation and proliferation of LUAD cells; significant induction of apoptosis and autophagy (non-cytoprotective). In vivo, suppresses lung adenocarcinoma tumor growth.	--	In vivo: 10 mg/kg 3×/week (mouse)	↓PAK1 (kinase linked to proliferation); ↑autophagy (non-cytoprotective); ↑apoptosis.	48–72 h (in vitro); 3 weeks (in vivo)	Not reported (no specific adverse effects reported)
[30]	Prostate (CRPC)	In vitro: LNCaP, C4-2, 22Rv1 (AR+ prostate); In vivo: 22Rv1 xenograft in castrated mice.	IVM 4–12 µM in culture (48 h); In vivo: 10 mg/kg i.p., 3×/week.	G0/G1 arrest, apoptosis, and DNA damage in CRPC cells. ↓AR (full-length and variants) and PSA; ↓E2F1 and AR signaling by FOXA1 blockade; ↑γH2AX (DSB). In vivo reduces tumor volume 22Rv1 (↓Ki67, PSA).	Enzalutamide trial—synergy (IVM IC50 ↓ with enzalutamide)	In vivo: 10 mg/kg 3×/week (mouse)	Direct target: FOXA1 and Ku70/Ku80 (DSB repair); affects AR/E2F1 signaling; ↑apoptotic cascade (PARP).	48 h (in vitro); 3–4 weeks (mice)	Preferential in AR+ cells (IC50 2–3 × lower in AR+ vs. AR−); no significant systemic toxicity reported.
[31]	Breast	In vitro: MCF-7 (ER+), MDA-MB-231 (triple-negative); Normal cells 184A1.	IVM 2.5–30 µM in culture (24 h).	Inhibits viability of MCF-7 (IC5024 µM) and MDA-MB-231 (IC5034 µM) much more than normal cells 184A1 (IC50~68 µM). ↑Dose-dependent apoptosis (AO/EB) in MCF-7/MDA-231. ↑ROS, ↑DNA damage (comet assay), MMP-Δψm in cancer cells.	--	–	Oxidative stress and mitochondrial apoptosis pathway (↑ROS, ↓glutathione); DNA damage (comet assay).	24 h	Normal IC50 >> tumor (better selectivity in cancer); low cytotoxicity in normal cells (collagen).
[10]	Pancreas (ductal)	In vitro: MiaPaCa-2, PANC-1 (pancreatic cells); ex vivo: patient organoids; In vivo: xenografts in mice.	IVM (± Gemcitabine 5 µM): IVM 2.5–10 µM (48–72 h); with gemcitabine 5 µM.	IVM + gemcitabine synergy: ↑proliferative inhibition (↓CI50). The combination induced G1 arrest (↓cyclins D1, ↓mTOR/STAT3) and ↑mitochondrial apoptosis (↑ROS, ↓mitochondrial Δψ). Decreases OCR and inhibits mitophagy. In vivo, IVM + gemcitabine suppresses tumorigenesis more than gemcitabine alone.	Gemcitabine (5 µM)—clear synergy	In vivo: (not specified)	↓mTOR/STAT3; G1 arrest (↓cyclins D1/CDK4); ↑ROS/↓mitochondrial Δψ; ↓mitophagy.	48–72 h (in vitro); 3 weeks (xenografts)	No notable adverse effects reported; IVM + gemcitabine better inhibition vs. gemcitabine alone (synergism).
[32]	Liver (hepatocellular carcinoma, HCC)	In vitro: HuH6, HepG2, SNU-182 (hepatocellular cancer); In vivo: HCC xenografts in mice.	IVM 5–20 µM in culture; In vivo: (dose details not specified)	IVM inhibited dose-dependent proliferation of HCC lines and ↑apoptosis. Inhibits migration, colonies, and CSC function. Suppresses oncogenic signaling mTOR/STAT3 and EMT and “stemness” markers. In mice, IVM reduced tumor formation and growth without apparent toxicity; also showed synergy with sorafenib.	Sorafenib—marked synergy (CI < 1)	–	↓mTOR/STAT3; ↓EMT pathway (↓E-cad, ↓N-cad); ↓stem markers (Nanog, c-Myc).	48–72 h (in vitro); 4–6 weeks (xenografts)	No systemic toxic effects in mice; IVM synergistic with sorafenib; global inhibition of essential oncogenic pathways.
[17]	High-grade serous carcinoma (ovarian cancer)	In vitro: Chemoresistant high-grade serous ovarian cancer cell lines: OVCAR8 and OVCAR8 PTX^RP (resistant to carboplatin/paclitaxel) and a non-tumoral line HOSE6.3 (normal ovarian epithelium).	Paclitaxel alone and in combination with repositioned drugs (pitavastatin, metformin, ivermectin, itraconazole, alendronate). Cells were exposed to Ivermectin in the range 0.39–50 μM (48 h), and pitavastatin in 0.04–5 μM (48 h). The combinations of Paclitaxel with each drug were administered simultaneously in a fixed ratio (0.25–4× IC_50_ of each drug).	Combination of Paclitaxel + Ivermectin or Paclitaxel + Pitavastatin produced maximum cytotoxicity and strong synergy in both chemoresistant lines, surpassing the effect of each drug alone.	--	Ivermectin was combined with Paclitaxel in fixed ratios (0.25–4× IC_50_). The combination Paclitaxel + Ivermectin (and similarly Paclitaxel + Pitavastatin) showed the highest synergy and antitumor effect. (Combinations with metformin, itraconazole, etc., were also tested with lower relative synergy)	Pitavastatin: 0.04–5 μM (48 h) Ivermectin: 0.39–50 μM (48 h) Paclitaxel: initial dose unknown, combined at 0.25–4× IC_50_ with each drug	No specific molecular targets were investigated in this experiment; it was assumed each drug acts via independent mechanisms (mutually exclusive model in synergy analysis).	48 h (continuous exposure)

**Table 4 pharmaceuticals-18-01459-t004:** Systematic Search Strategies for identified documents on the application IVM in Oncology.

Topic	Keywords Boolean Search
IVM in cancer treatments	(ivermectin AND this AND application AND in AND cancer AND treatment).
Chemical Properties of Ivermectin	(((Ivermectin Physicochemical Properties *) AND (stability *) AND (solubility *)) OR (chemical family)) OR ((physical properties *) OR (chemical properties)) OR ((biological activity *)
Oncological Properties of Ivermectin	((cancer * OR signaling in cancer cells AND morphologic AND feature *) AND (receptors AND enzymes AND metabolic pathways *)))

The symbol (*) represents a wildcard to help in the search when a word has multiple spelling variations.

**Table 5 pharmaceuticals-18-01459-t005:** Units of analysis for bibliometric analysis in VOSviewer.

	Analysis Unit
Co-authorship	Authors
	Organizations
	Countries
Co-occurrence	Keywords
	Author Keywords
	Index Keywords
Citation	Documents
	Sources
	Authors
	Organizations
	Countries
Bibliographic Coupling	Documents
	Sources
	Authors
	Organizations
	Countries
Co-citation	Cited references
	Cited sources
	Cited authors

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
