# Peer review of "Ivermectin as an Alternative Anticancer Agent: A Review of Its Chemical Properties and Therapeutic Potential"

_pharmaceuticals, 2025, doi:10.3390/ph18101459_

Round 1

Reviewer 1 Report

Comments and Suggestions for Authors

This article is a literature review of the anthelmintic drug ivermectin (IVM) and its alternative uses in oncology. In particular, the authors focus on IVM’s chemical properties and target pathways. This review may provide useful information for repurposing IVM in clinical oncology. However, the article lacks proper citations and figures referenced in the text as well as sufficient explanations in the Results section. The Discussion and Conclusions sections contain unnecessary redundancy. The label for Figure 7 is written in a language other than English. Therefore, this reviewer was unable to properly assess the manuscript.

Author Response

This article is a literature review of the anthelmintic drug ivermectin (IVM) and its alternative uses in oncology. In particular, the authors focus on IVM’s chemical properties and target pathways.

 This review may provide useful information for repurposing IVM in clinical oncology.

Thank you for the comment!

However, the article lacks proper citations and figures referenced in the text as well as sufficient explanations in the Results section.

We thank the reviewer for this observation. In the revised manuscript, we have added the missing citations and figures referenced in the text, and we have expanded the Results section with clearer explanations to ensure that the findings are fully supported and better contextualized.

The Discussion and Conclusions sections contain unnecessary redundancy.

We appreciate the reviewer’s observation regarding redundancy between the Discussion and Conclusions sections. In the revised manuscript, we have carefully streamlined both sections to ensure that the Discussion focuses on interpreting and critically analyzing the findings in relation to existing literature, while the Conclusions provide only a concise synthesis of the main contributions, limitations, and directions for future research. This revision eliminates overlap and improves the clarity and structure of the manuscript.

The label for Figure 7 is written in a language other than English. Therefore, this reviewer was unable to properly assess the manuscript.

We thank the reviewer for pointing this out. In the revised version of the manuscript, the label for Figure 7 has been translated into English to ensure clarity and consistency throughout the paper. We have carefully reviewed all figures and tables to confirm that their labels, legends, and captions are presented in English, allowing for proper assessment of the manuscript.

Reviewer 2 Report

Comments and Suggestions for Authors

Efforts are being made to find more effective and less toxic therapeutic alternatives in cancer treatment. In this context, drug repurposing is especially gaining attention as a promising strategy. Ivermectin (IVM), which was originally discovered as an antiparasitic agent in the 1970s, has been recently used as an anti-cancer drug, although most physicians have cautioned about the lack of clinical evidence supporting its use in cancer treatment.

In this work, Naula et al. conducted a comprehensive literature review to evaluate IVM’s chemical characteristics and assess its applicability as an alternative therapeutic strategy in oncology. By summarizing the findings in 37 studies, which passed their Inclusion/Exclusion criteria, the authors highlighted the physicochemical profiles of IVM. The authors also revealed that clinical validation of IVM’s use in cancer treatment remains limited.

The methods used in the systematic review are also clearly described in the manuscript. I believe that this study is valuable to the field, especially because it provides a foundational framework for future research on IVM’s drug repurposing for cancer treatment, and that it can be published in its current form.

However, I have some minor comments that should be addressed before publication below;

  1. Tables 3 and 4 are very difficult to read, the authors should improve the formatting.

  1. Is there any reason to use Spanish for the X-axis’ labeling (cancer type)?

Author Response

Efforts are being made to find more effective and less toxic therapeutic alternatives in cancer treatment. In this context, drug repurposing is especially gaining attention as a promising strategy. Ivermectin (IVM), which was originally discovered as an antiparasitic agent in the 1970s, has been recently used as an anti-cancer drug, although most physicians have cautioned about the lack of clinical evidence supporting its use in cancer treatment.

In this work, Naula et al. conducted a comprehensive literature review to evaluate IVM’s chemical characteristics and assess its applicability as an alternative therapeutic strategy in oncology. By summarizing the findings in 37 studies, which passed their Inclusion/Exclusion criteria, the authors highlighted the physicochemical profiles of IVM. The authors also revealed that clinical validation of IVM’s use in cancer treatment remains limited.

The methods used in the systematic review are also clearly described in the manuscript. I believe that this study is valuable to the field, especially because it provides a foundational framework for future research on IVM’s drug repurposing for cancer treatment, and that it can be published in its current form.

Thank you for the comment!

However, I have some minor comments that should be addressed before publication below;

  1. Tables 3 and 4 are very difficult to read, the authors should improve the formatting.

We thank the reviewer for these helpful observations. In the revised version of the manuscript, we have reformatted Tables 3 and 4 to improve readability by adjusting column alignment, and spacing, by simplifying the layout for clearer presentation of the data.

2. Is there any reason to use Spanish for the X-axis’ labeling (cancer type)?

Regarding the use of spanish on the X-axis labeling, we acknowledge this oversight. All axis labels, figure captions, and legends have now been translated into english to ensure consistency and accessibility for an international readership.

Reviewer 3 Report

Comments and Suggestions for Authors

The manuscript "Ivermectin as an Alternative Anticancer Agent: A Review of Its Chemical Properties and Therapeutic Potential" lists the chemical properties, biological potential, and repurposing of Ivermectin as an anticancer agent. The manuscript is well written, and the data are organized scientifically. However, some minor concerns need to be addressed. 

  1. The introduction is divided into too my paragraphs. The introduction needs to be revised for streamlining and to address the novelty of the manuscript. 
  2. Figure 2 is unclear, and text formatting is required for better clarity. It is also applicable to other figures. 
  3. Explain the legend of Figure 7 for more clarity and better understanding. 
  4. Discussion is also too much paragraphed and needs to be streamlined.  
  5. The manuscript needs to be revised for minor typos and grammatical errors. 
  6. Some of the references are incomplete, such as ref 13, 17, etc.

Author Response

The manuscript "Ivermectin as an Alternative Anticancer Agent: A Review of Its Chemical Properties and Therapeutic Potential" lists the chemical properties, biological potential, and repurposing of Ivermectin as an anticancer agent. The manuscript is well written, and the data are organized scientifically. However, some minor concerns need to be addressed. 

  1. The introduction is divided into too my paragraphs. The introduction needs to be revised for streamlining and to address the novelty of the manuscript. 

We sincerely appreciate your valuable comments on our manuscript and your suggestions for improving the introduction. We have carefully considered your observations regarding the number of paragraphs and the need to streamline the text to highlight the study's novelty.

We have restructured the introduction to enhance its fluidity and conciseness.

To explicitly highlight the novelty of our research, we have incorporated a specific section in the revised introduction that underscores the unique contribution of this manuscript.

We thank you again for your time and thorough review, which have undoubtedly contributed to improving the quality of our work.

2. Figure 2 is unclear, and text formatting is required for better clarity. It is also applicable to other figures. 

We thank the reviewer for this observation. In the revised manuscript, Figure 2 has been redrawn with improved resolution, simplified labeling, and clearer text formatting to enhance readability. e have also applied the same formatting improvements to the other figures.

3.Explain the legend of Figure 7 for more clarity and better understanding. 

We appreciate the reviewer’s suggestion. The legend of Figure 7 has been revised and expanded to provide a clearer explanation of the variables, symbols, and abbreviations used. Additional contextual information has been included to guide the reader in interpreting the figure accurately and to strengthen its contribution to the Results section.

4.Discussion is also too much paragraphed and needs to be streamlined.  

We appreciate the reviewer’s observation. In the revised manuscript, we have improved the discussion section to ensure that it focuses on interpreting and critically analyzing the findings concerning existing literature, and it was streamlined. 

5.The manuscript needs to be revised for minor typos and grammatical errors. 

We appreciate the reviewer’s observation. The manuscript has been carefully revised to correct minor typographical and grammatical errors. We also conducted an additional language check to ensure clarity, accuracy, and consistency throughout the text.

6.Some of the references are incomplete, such as ref 13, 17, etc.

We appreciate the reviewer’s observation. The incomplete references (e.g., ref. 13, 17) have been carefully revised and updated with full bibliographic details in the reference list to ensure accuracy and consistency.

Round 2

Reviewer 1 Report

Comments and Suggestions for Authors

I recognized the revision to Figure 7. However, I do not believe the Results section has improved enough for publication.   

Author Response

We appreciate the reviewer’s observation .Tables 3-5, were remaking in horizontal way , easy to read.

All references were carefully reviewed, to eliminated redundance and unnecessary cites.

We provide a new version with Highlight any revisions to the manuscript, so editors and reviewers can see any changes made.

We provide a cover letter to respond to the reviewers’ comments and
explain, point by point, the details of the manuscript revisions.